# Evaluation of the Dissemination of the South African 24-Hour Movement Guidelines for Birth to 5 Years

**DOI:** 10.3390/ijerph18063071

**Published:** 2021-03-17

**Authors:** Catherine E. Draper, Takana M. Silubonde, Gudani Mukoma, Esther M. F. van Sluijs

**Affiliations:** 1SA MRC Developmental Pathways for Health Research Unit, School of Clinical Medicine, University of the Witwatersrand, 2050 Johannesburg, South Africa; catherine.draper@wits.ac.za (C.E.D.); takanawill@gmail.com (T.M.S.); mukomagudani@gmail.com (G.M.); 2Health through Physical Activity, Lifestyle and Sport Research Centre & Division of Exercise Science and Sports Medicine, Department of Human Biology, Faculty of Health Sciences, University of Cape Town, 7700 Cape Town, South Africa; 3MRC Epidemiology Unit & UKCRC Centre for Diet and Activity Research (CEDAR), School of Clinical Medicine, University of Cambridge, Cambridge CB2 0QQ, UK

**Keywords:** movement behaviour guidelines, implementation, low- and middle-income country

## Abstract

South Africa (SA) launched their 24-h movement guidelines for birth to five years in December 2018. The guideline dissemination plan adopted a “train-the-trainer” strategy through dissemination workshops with community-based organisations (CBOs) working in early childhood development. The aim of this paper is to: (1) document this dissemination process; and (2) report on the feasibility of implementing the dissemination workshops, the acceptability of the workshops (and guidelines) for different end-user groups, and the extent to which CBO representatives disseminated the guidelines to end-users. Fifteen workshops were held in seven of SA’s nine provinces with a total of 323 attendees. Quantitative and qualitative findings (*n* = 281) indicate that these workshops were feasible for community-based dissemination of the guidelines and that this method of dissemination was acceptable to CBOs and end-users. Findings from follow-up focus groups (6 groups, *n* = 28 participants) indicate that the guidelines were shared with end-users of CBOs who participated in the focus groups. An additional musical storytelling resource, the “Woza, Mntwana” song, was well-received by participants; sharing via WhatsApp was believed to be the most effective way to disseminate this song. These findings confirm the feasibility and acceptability of culturally appropriate and context-specific community-based dissemination of behavioural guidelines in low-income settings.

## 1. Introduction

Guidelines on young children’s movement behaviour have been developed by a number of countries in recent years, including Canada [1], Australia [2], United Kingdom (UK) [3], and New Zealand [4]. Global guidelines on these behaviours were launched by the World Health Organization in 2019 [5]. Dissemination of these guidelines has been varied and depends to a large extent on how the guidelines were commissioned and funded. In Australia, the UK, and New Zealand, the development of guidelines was led by relevant government departments, and these departments have then taken control of dissemination with advice from scientific advisors.

South Africa (SA) is the first low- and middle-income country (LMIC) to develop 24-h movement guidelines for birth to five years, launched in December 2018 [6]. The development included stakeholder consultation [7] to ensure that the guidelines were contextually relevant and appropriate for SA in general and low-income settings in particular. While led by academics, the guideline consensus panel had representation from a range of sectors (including health, education, sport, and early childhood development) and received funding from a not-for-profit organisation (the Laureus Sport for Good Foundation). The guidelines were intentionally not “owned” by any sector, leaving a fair degree of autonomy when it came to planning for guideline dissemination beyond the official launch event.

The SA guidelines’ preamble states that “these guidelines are for those who have an interest in the health and development of all children from birth to five years, including parents and family, educators, caregivers, health professionals, and community workers. These guidelines should be implemented in homes, early childhood development programmes and centres, or any setting where children may engage in these movement behaviours” [6]. Given the wide range of intended end-users, dissemination would not be able to rely solely on online methods; any dissemination strategies would need to be optimised for low-income settings. Although SA is one of the biggest adopters of mobile technology in Africa, mobile data costs remain high [8] and access to computers and internet remains a challenge in SA due to the “Digital Divide” [9]. However, WhatsApp is a popular, accessible, and cost-effective digital platform for dissemination in SA and is used extensively by the Department of Health (for examples: https://www.praekelt.org (Access date: 13 January 2021)). For example, recent guidance on the use of cloth face masks for preventing the spread of COVID-19 [10] was disseminated through numerous infographics (http://www.health.gov.za/index.php/component/phocadownload/category/631 (accessed on 13 January 2021)). The movement guidelines [6] therefore also included the production of graphics (in 11 official SA languages) that could be easily shared on WhatsApp.

Apart from this mobile solution, there was limited precedent to draw on for the dissemination of health promotion guidelines in SA. The best known health promotion guidelines are the SA food-based dietary guidelines [11], but while these are widely supported by the Department of Health, dissemination at community level is limited [12]. Although the COVID-19 infographics have been widely distributed on social media and are easy to share on WhatsApp, their reach has not yet been formally evaluated.

Despite having little to draw on in terms of health promotion guideline dissemination, the consensus panel agreed that community-based dissemination of the movement guidelines was a priority. In light of the number of community-based organisations (CBOs) in the early childhood development (ECD) sector in SA (a sector well represented on the panel and in the stakeholder consultation process), there was agreement to focus dissemination efforts at working with these CBOs to get these guidelines into the hands of the intended end-users. Specifically, this meant making contact with CBOs working with caregivers of young children and those working with 0–5-year-old children in early care and education settings (referred to as ECD practitioners in SA). A “train-the-trainer” approach was adopted, such that CBO representatives would be invited to attend a local workshop so that they could be “trained” on how to use the guidelines. In turn, these CBO representatives could then pass on their knowledge and understanding about the guidelines to their staff, who could share the guidelines with the caregivers and/or ECD practitioners (end-users) with whom they worked. Importantly, these workshops would also be an opportunity to share printed resources, specifically the guidelines infographic [6], shown in Figure 1, which is available in all of the 11 official SA languages.

Stakeholder consultation supported these decisions regarding these dissemination workshops [7]. Firstly, face-to-face workshops were considered essential to maximise understanding, given the novelty of these guidelines, the limited awareness about recommendations for young children’s movement behaviours, low levels of literacy, and other contextual challenges in low-income SA communities. Secondly, these workshops would likely be well received, given end-users’ positive perceptions of the guidelines and their eagerness to learn about these guidelines [7]. Lastly, these guidelines should be framed in the context of ECD more broadly, rather than just in terms of health. The need for this framing has been identified in other intervention development work with this age group in SA [13,14].

There is limited published evidence on the feasibility, acceptability, and effectiveness of movement guideline dissemination strategies, and this evidence focuses on adults Ref [15]. This paper therefore aims to: (1) document the dissemination process of the SA 24-h movement guidelines for birth to five years, including the development of additional creative materials; and (2) report on:The feasibility of implementing the dissemination workshops;The acceptability of the dissemination workshops (and the guidelines) for different end-user groups; andThe extent to which CBO representatives disseminated the guidelines to their staff and end-users.

## 2. Materials and Methods

Ethical approval for this mixed-methods study was obtained from the University of the Witwatersrand’s Human Research Ethics Committee (Medical; ref: M190416). All participants gave written, informed consent for their involvement in the study. This project had four phases, which took place from April 2019—November 2020, as shown in Figure 2.

### 2.1. Programme Theory

Figure 3 presents the programme theory for the dissemination of the SA 24-h movement guidelines for birth to five years. This programme theory outlines the underlying rationale for and purpose of the dissemination workshops; the intended short-, medium-, and long-term outcomes of the workshops; and the conditions necessary for the workshop outcomes to be achieved.

### 2.2. Phase 1: Compilation of CBO Database and Dissemination Workshop Planning

Phase 1 comprised desk-based research to identify CBOs who work with caregivers of 0–5-year-old children and/or ECD practitioners across all nine provinces of SA. This also included national ECD organisations and networks in order to obtain a nationally representative list of potential contacts who may be interested in attending a guideline dissemination workshop. Organisations represented on the guideline consensus panel assisted with sharing their ECD contact lists, as a starting point, and the list was added to in a “snowball” manner. As this list was compiled, it became evident that individual ECD centres (such as preschools) were included in some contact lists, and although these did not fit the original approach of working just with CBOs, we opted to be inclusive and have as many interested ECD CBOs and centres as possible to maximise the reach of our dissemination efforts.

A dedicated email address was set up (ecdmovementguidelines@gmail.com), and all CBO and network contacts on the list (*n* = 118) were emailed to determine their level of interest in this project and their willingness to attend a dissemination workshop in their region. A dedicated mobile number was set up so that contact could be maintained via WhatsApp as well. While we were willing to hold workshops across as many provincial locations as possible, workshops were ultimately planned for locations where positive responses were received from individuals and CBOs on the contact list. A formal invitation was sent to all contacts for the workshop in their city or location, and they were asked to distribute it to their organisation, colleagues, and networks. Workshops were planned to be conducted over two hours, which was believed to be a feasible length for attendees. While a longer workshop would have enabled further discussion and evaluation activities, it was anticipated that this would have led to lower levels of attendance due to busy schedules. The start and end times were included on each invitation and also mentioned that light refreshments would be served (generally an expectation for community-based workshops) and that attendance was free.

Workshop venues were arranged with CBOs where possible (accessible and community-based) or through existing contacts at other higher education or research institution venues through existing contacts. Where workshops were held at academic institutions, students and academics were also invited to attend. One workshop was held at the offices of the provincial government (Education). In alignment with our inclusive approach to maximise reach, where workshops were requested for end-users or any other stakeholders (as opposed to CBO representatives), these requests were accommodated.

### 2.3. Phase 2: Implementation and Evaluation of Dissemination Workshops

Fifteen workshops were arranged in seven of SA’s provinces, covering both urban and rural settings, to take place in September 2019. One researcher facilitated 14 of the workshops; the other was facilitated by a second. Printed copies of the guideline infographic (in the language appropriate for the setting) were arranged for distribution at the workshops. The format of the workshop is outlined in Table 1.

A short evaluation questionnaire was developed to obtain attendees’ feedback, to be completed at the end of the workshop (see Appendix A for discussion questions). Time was also allocated at the end of the workshop to have a group discussion with attendees, led by the workshop facilitator, in order to obtain additional qualitative feedback from attendees. These discussions could not accurately be described as focus groups, especially since the size of some groups exceeded what would be recommended for meaningful and in-depth discussion. Nevertheless, similar to focus group discussions, they were intended to draw on the social dynamics of the groups and provide an opportunity for a facilitated, stimulating, and interactive exchange between attendees.

Questionnaire responses were entered into Google Forms and exported as an Excel spreadsheet for analysis. Frequencies were calculated for questionnaire responses in Excel and are presented as percentages of responses who responded “strongly agree”, “agree”, “neutral”, “disagree”, or “strongly disagree”. Open-ended questionnaire responses were grouped according to content, and group discussions were recorded transcribed verbatim. These open-ended responses and transcripts were thematically analysed by the lead author using a largely deductive approach [16]. The group discussion questions formed the basis of an initial thematic framework, and this was further developed to encompass the following main themes: (1) perceptions of the workshops, (2) the role of parents (and caregivers), (3) challenges to implementation, (4) suggestions for the workshops. These themes relate to the feasibility and acceptability of the workshops and provide insight into the acceptability of the guidelines and their implementation. After the initial stage of familiarisation with the data, codes were generated based on these themes. The next step involved searching for themes in the transcripts, and once coded sections of text were summarised for each theme, illustrative quotes for each theme were extracted.

### 2.4. Phase 3: Marketing Campaign Development

In August 2019, contact was made with a marketing and creative content development company, *Creatrix* (www.creatrix.co.za (accessed on 13 January 2021)) at a “community of practice” workshop. *Creatrix* was contracted to develop a marketing campaign name for the guidelines and to produce a guidelines song. This song would draw on the concepts of musical storytelling, which is a method that draws on the power of music and the spoken word to disseminate content. We anticipated that this would be appealing for ECD practitioners and caregivers of young children (and to young children as well) and would build on SA’s rich culture of music and oral storytelling. Furthermore, we expected that this would be especially relevant in low-income communities where the acceptability and utility of printed resources can sometimes be limited, due to low levels of literacy and education. The development of this campaign (and song) took place after the dissemination workshops had been implemented and was informed by the workshop evaluation findings.

### 2.5. Phase 4: Follow-Up Focus Groups

The purpose of these follow-up focus groups with CBO representatives (who work with caregivers of young children and/or ECD practitioners) was to evaluate the extent to which workshop attendees disseminated the guidelines to their CBO staff and end-users, since it would not have been feasible to follow-up with all end-users associated with the CBOs and workshop attendees. Not all focus group participants had necessarily attended a dissemination workshop but had at least been made aware of the guidelines from a CBO representative who had attended a workshop. The focus groups also obtained feedback on the marketing campaign and song. Focus group guide questions are provided in Appendix A.

Focus groups (6 groups, *n* = 28 participants) were scheduled to be held face-to-face starting in March 2020, but only one the focus group in Cape Town (*n* = 3) could take place in person before COVID-19 lockdown restrictions in SA prohibited inter-provincial travel for a number of months. A further five focus groups were conducted online in July 2020 using a range of online platforms, including Zoom, Skype, and WhatsApp group calls. Where appropriate, focus group participants were sent mobile data to allow them to connect online. It was necessary to be flexible with the platform used based on mobile connectivity issues in many low-income settings and the devices available to participants. Participants were from CBOs in KwaZulu-Natal (KZN) Province (rural, *n* = 9) and Gauteng Province (urban and peri-urban, *n* = 16). These focus group discussions were recorded and transcribed verbatim; a similar analytical approach was taken to identify the key themes from these focus group discussions, which included: (1) feedback on the workshops, (2) extent of dissemination to CBO staff and end-users, (3) responses of CBO staff and end-users, and (4) perceptions of “Woza, Mntwana” song and recommendations for its dissemination.

## 3. Results

### 3.1. Implementation of Dissemination Workshops

Details of the workshop locations and attendees are provided in Table 2. Attendees not being able to stay for the evaluation component of the workshop accounted for the discrepancy in attendees and evaluation forms completed. The format of workshop 6 was different to other workshops; the team was provided with approximately 30 min to present the guidelines at a community event (held at a church) that was addressing a range of health and social issues. The presentation of the guidelines was therefore required to be more didactic than in other workshops, and while those present (approximately 120 community members) were able to ask questions and comment on the guidelines, it was not feasible to obtain extensive feedback or have evaluation forms completed. Workshop 8 also had a large audience, which also necessitated a more didactic approach. However, this workshop took place in a well-appointed lecture venue at a university, which was conducive for interaction between the facilitator and attendees.

### 3.2. Quantitative Evaluation of Workshops

Workshop attendees’ responses to the evaluation questions are presented in Table 3. When asked which guidelines participants thought were the most important, 70.5% selected “All of them”, 21.0% physical activity, 3.6% screen time, 2.1% sleep, and 0.7% sitting time (*n* = 275). Workshop attendees’ written feedback was generally positive about the workshop and the guidelines, indicating that they learnt something and would share the information. A few attendees highlighted the need to involve parents and caregivers in these workshops, and some requested more practical examples in the workshop and that there should be more workshops in the future. Selected open-ended responses are included in Table 3.

### 3.3. Qualitative Evaluation of the Workshops

This section presents the qualitative feedback from workshop participants, based on the facilitated discussions that took place at the end of the workshops. Certain feedback from participants highlighted their positive perceptions of the guidelines and their recognition of the importance of the guidelines. However, while these perceptions are valuable to know, this section focuses on the feedback that relates directly to the aims of this evaluation. These qualitative findings support the quantitative findings regarding participants’ positive perceptions of the workshops and provide more detail regarding challenges to implementation. These relate to the need for resources and support highlighted in participants’ questionnaire responses, which were less positive than their responses about the workshops.

#### 3.3.1. Perceptions of the Workshops

The feedback from attendees about the workshops was generally positive and was mostly to do with the learning and understand they felt they gained by attending the workshop, which they believed could be communicated to parents, caregivers, and communities. Some attendees described the workshop as an “eye-opener”, and they spoke about how it helped them to understand guidelines in a simple way in terms of what children need and what is important and how to apply this. Attendees mentioned that the workshop gave them ideas of what to do with children in their care, including suggestions of making toys with materials at home and interacting with young children. The knowledge of what to do or what not to do was described as helpful, especially time recommendations for the age groups and behaviour, as well as the kinds of activities to do with the different age groups. For some attendees, the workshop was a good reminder of some of the things they already knew, and others spoke about learning things for themselves as parents or grandparents of young children. At some workshops, attendees found it to be a valuable opportunity to link with others who work in the field to share ideas and potentially share resources.


*It showed us a lot on what they need and what is important and what not to do, and how to explain to the parents like what not to do. (Bloemfontein)*

*This workshop helped me to understand a child who is aged from 0–1 years old, that from that age what a child is supposed to be doing and the activities that they should be taking part in. Then they move from 1–2 years old…I didn’t know that they mustn’t watch TV because when they don’t watch TV they have a higher concentration span. So it helped to get information like that and to even know how a child from 3–5 acts. (Giyani)*

*Another thing that I’ve learned from the workshop we need to have a proper talk with our grandmothers because most of the time our children especially at the early age they are spending the time with their grandmothers especially while you are working. So, we need to share the information with them so they can know what is expected from them and what they are doing. It is going to affect the life of the child. I’m happy about this workshop. (Mbombela)*

*I have learnt more because the training has helped me to see and learn a lot. I have seen and heard a lot of things like improvising, I cannot sit with children in our centre and say we don’t have money for toys. We can involve our parents to come to the centre and make these toys for our children with our own hands, home-made toys. (Polokwane)*

*So from my side I think you know it helps me to understand that technology is not the only way to make the children learn and move, because sometimes you think that if we have those tablets we put or some children with TV on their desk they watch the TV the whole day, they are learning while they do a lot of activities. (Pretoria)*


#### 3.3.2. The Role of Parents

The critical role of parents in implementing the guidelines was frequently discussed by attendees. It was seen as essential to get them on board and for them to understand the guidelines and to be consistent in implementing them. Attendees mentioned that in low-income settings, parents need support and empowerment for interacting with their children to stimulate their development (especially single parents) and to reduce the “gap between the parent and child”; they felt it was important that parents understand that it is not just up to the preschool to stimulate development of children.

For parents, it was acknowledged that parenting has changed and that children are more frequently making decisions about their behaviours (as they get older) and that families are more child-centred, with the consequence that the “child is the one that is leading the home”. However, this was seen to be detrimental to children, and guidance for parents about how to manage their children’s behaviours is essential. Added to this, it was mentioned that parents often have limited time with their children at home and that it is important to promote opportunities for interaction and connection between parents and their young children. Attendees maintained that through playing with their children, parents can better understand how they are developing and see where they are lacking. Some attendees highlighted the importance of parents spending more time with their children and not be “selfish with their time”. Although this was in contrast to the “hovercraft parenting” mentioned in one workshop, which related more to physical activity in terms of an overemphasis on safety and reducing children’s activity and not allowing them to take any risks that could stimulate their development.

Parents featured strongly in discussions about screen time. Attendees’ comments indicated that parents need more guidance on their children’s screen time, owing to a few reasons: times have changed, and these parents did not grow up with screens, so they do not have guidance to refer to; and there has been a significant increase in the availability and use of screens, greater access to free wi-fi. Attendees also discussed how screens were used by parents (often when they are busy) to keep children occupied and entertained, described as a “lazy parenting technique” which creates problems further down the line when parents are reliant on screen time to get their children to cooperate. Some attendees noted that screen time limits should not be up to the child and that parents need to “put their foot down”. In addition, attendees mentioned that children observed and emulated their parents’ screen behaviours, which were often not healthy, and that parents should be setting the right example with their own behaviours—particularly for screen time, but applicable to the other behaviours as well.


*You mentioned that restricting children with screen time especially. If we set a good example of not being on our phones the whole day, or in front of the television all day. That example is what the child remembers as well, it’s up to us to set good example especially for screen time, in this day and age where everyone has a phone and everyone has a television on the whole day we actually set an example for that. (Pretoria)*


#### 3.3.3. Challenges to Implementation

Although attendees were generally positive about the guidelines, they mentioned a number of challenges to implementation of the guidelines. Some of these relate to what has been mentioned already regarding parents and ECD practitioners but speak to broader social challenges. A key challenge is behaviour change that confronts how people have been brought up, especially if they are too busy and do not have the time, space, and/or resources to make these changes. Attendees mentioned that ECD practitioners in some settings have low levels of enthusiasm, energy, and motivation and are not willing to work hard. Other ECD practitioners in low-income settings were said to have a reliance on resources, always complaining about a lack of resources and not being able to think of alternatives without resources. Many ECD practitioners have a large number of children to care for, which can exacerbate these challenges. These challenges make it unlikely that the guidelines would be implemented by these types of practitioners in these settings.


*So think one of the biggest…challenges of implementing something new is the idea of behaviour change and trying to influence that is quite difficult. In our setting it is sort of easy to revert back to that, oh we don’t have the time, we don’t have the space…I don’t have the money to be able to do this. (Cape Town)*

*Unfortunately, everyone is busy these days. You go to the clinic and hospitals and they tell you that they are too busy for that. You go to a pre-school where they have a full class of kids and they don’t know what to do with them. So everyone is busy, the parents are busy, they wake up at 3 am to go to work, but the least they can do in dedicate a few minutes of their time to their child. (Johannesburg)*


Parents not being at home because they are both working is another challenge (across income settings), as it limits time spent with children, and there are many “easy” options, such as screen time and take-away food. Attendees mentioned the issue of young mothers (particularly in low-income settings) who are absent and do not know what to do with their children, rather relying on grandparents to look after their children. Broader social (and related) challenges that were highlighted by attendees were safety, poverty and financial constraints, and poor nutrition. Safety was specifically mentioned with respect to keeping children disciplined, safe, and secure. This was understood to potentially limit children’s movement if it was perceived to be dangerous to play outside.


*I think certainly with ECD practitioners and caregivers working with your children they sometimes feel like they have to limit what the children are doing because it is more about keeping them disciplined and safe and it is often for good reasons like safety and security and that sort of thing. (Cape Town)*

*There is a lot of factors that influence this 24-h day…where we work, the number one factor is nutrition and I think that we need to factor that in when we think about the 24-h day. There are also a lot of social challenges, there is a lot of poverty and where there is poverty, the kids are not eating there is malnutrition. (Cape Town)*


Other challenges mentioned by attendees included the need for assistance in rural areas, the limited funds available from government, transport challenges, and resources for ECD centres and community-based organisations.


*And funding and mentoring after the workshop, you can mentor us until you see the things happening and funding, funding is a key problem. (Sweetwaters)*


#### 3.3.4. Suggestions for the Workshops

The main suggestion relating to the workshops was to have more of them and to do follow-up workshops. Attendees also asked for workshops for other groups that were possibly not well represented at the workshop they attended, for example, for parents, grandparents, ECD practitioners who work with the different age groups, and community leaders. Where the workshop facilitators were asked to return to a specific group to do workshops, the question was posed about whether those who had attended could facilitate future workshops. However, attendees in these groups did not seem to feel empowered to do the workshops themselves, and they mentioned that parents and practitioners would be more inclined to listen to the original workshop facilitators. This suggests that the “train-the-trainer” model may not be feasible in all settings.


*Because we as the teachers would be giving them a brief workshop here but the problem is the parents and they always have the questions that need to be answered. So, my plea is not let it be a once-off thing. Yes, we hear it but when you get to the centre and stand in front of them there is some information that you forget about. So, please we hope to see you next time again. (Mbombela)*


Requests were also made to cover other related topics, such as developmental milestones, nutrition, and discipline; and to do workshops in rural areas (requested by those in urban areas) where they believed was a need for training of ECD practitioners and parents. Attendees requested more practical examples, ideas, and activities, especially for the different age groups, to be included in the workshops. While some attendees requested more resources, such as toys, puzzles, and shapes, other attendees spoke about how recycled materials can be used to make toys, to reduce the reliance on resources being provided.


*What I think what would be great to maybe even add to this because obviously you weren’t able to add all the details and some more extensive ideas. Like what we would do with those activities, what we would do in that programme is that we have a ball, the one day you let the children roll it, and the next day do something different with it. You will teach the caregivers something done in a different way and these are the 20 different things that you can do with it. And some more practical ideas like that might help and also in infographic form and even in this type of form as well. (Sweetwaters)*


Lastly, suggestions were made to use more creative means to disseminate the guidelines, such as musical storytelling, and to provide additional information about the benefits of meeting the guidelines and the consequences of not meeting the guidelines, since this was not always understood or obvious.

### 3.4. “Woza, Mntwana” Campaign

The name of the campaign developed by *Creatrix* is “Woza, Mntwana”, which means “come here, child” in isiZulu, one of the most commonly spoken official languages in SA. The “Woza, Mntwana” logo is shown in Figure 4. “Woza, Mntwana” represents the desire to create connection between young children and their caregivers and to engage in behaviours that promote their health and development. This aligns with the nurturing care framework [17], as it encourages responsiveness, stimulation, and healthy behaviours. The logo was created from elements and shapes that are used in the guideline infographic (Figure 1), also using the colours of the SA flag. The overlapping colours used to create the letters are reminiscent of arms, legs, heads, and cartoon like “movement” lines of emphasis.

The lyrics of the “Woza, Mntwana” song are provided as Figure 5, and a video of the song, which has simple and fun dance moves for children to follow, is available on YouTube: https://www.youtube.com/watch?v=E11vCWPhxNk&t=37s (accessed on 13 January 2021) (released November 2020; available as Appendix A). This more recent video will be disseminated in early 2021 to CBOs on the database created as part of Phase 1 of the project.

### 3.5. Follow-Up Focus Groups

#### 3.5.1. Feedback on Workshops

At the follow-up focus groups, participants’ feedback about the workshops (and the guidelines) continued to be positive, and the importance of the screen time guideline for young children was emphasised again.


*I would say at first, they were like laughing…”what is happening with the screen?”…then until we explained to them that the screen is not good for the child, because the screen takes too much time of the child. And that is when they had to understand that okay, maybe that is why some of the things…At first, we didn’t know…So if you explain it better to the parents, then the parents tend to understand. (Gauteng, peri-urban 1)*

*I think screen time affects the entire daily routine because as I have been saying the children watch TV till late and this affects them in preschool the next day so I am really keen to give this information to the teachers so they can relay it to the parents. (KZN 1)*

*The workshop really helped us in terms of strengthening our knowledge, especially around physical movement because we were looking at the milestones of the children. So, it would help us to educate our parents and our caregivers, especially around physical activity for the child around, movement, around sleep and around screen time…So, it really helped us from a parenting perspective so that we can empower our caregivers or our beneficiaries that they can also do those activities during the day with their children as well. (Gauteng, peri-urban 1)*

*We’d like to just sincerely thank you for the knowledge that you gave us, and it has been helpful for us in through our work and with the clients that we are working with, and building as well the relationship between the mother and us, so that they trust us more with their children, and they trust us more that we’re going to help them in so many ways; not just…in a health related way, but also in a way of bonding with their children and learning the child’s stimulation and the milestone for their children. So, we really appreciate that. (Gauteng, peri-urban 1)*


#### 3.5.2. Extent of Dissemination to CBO Staff and End-Users

Participants reported on how they had shared the guidelines with staff who had not attended the dissemination workshop in their region, and one CBO spoke about how they had shared the guidelines with their staff even before the workshop, but the workshop (and receiving the guidelines) helped even more with incorporating the guidelines into their work.


*When it had just came out I think it was through (ECD network name), we heard about it and we were so excited. And we did a…training morning about it and it was quite interesting for us to us to see the numbers and we played a little bit of a guessing games on “how much do you think they can sit?” And it was very interesting for the team to see what it was, so we were already enthusiastic about it, and then we managed to attend that meeting…that was also helpful. And then what we did with it was that, in the homes we already have a curriculum where we do different things every week so movement has always been part of what we did but we could add more from of your ideas, and we could change one whole lesson from our curriculum so it could be based around the movement guideline. During that lesson we were able to hand out the pamphlets and things so there was that extra focus and then we could remind them that when we talk about TV time or sitting time then it would link back to these movement guidelines. (KZN)*


Participants spoke about how they had incorporated the guidelines and what they learnt at the dissemination workshops into their work with end-users. This included activities such as workshops with parents and caregivers (particularly where they were already encouraging stimulating interactions between caregivers and young children to promote development); training with ECD practitioners, “play group” leaders and other partners; and campaigns, such as mental health awareness month. They were also able to share the printed resources they received at the dissemination workshops with these end-users and other stakeholders in their networks, including government. CBOs who work with health facilities were able to share the guidelines with healthcare staff and community health workers and were able to include the guidelines in their health screening of young children.


*We shared with the team at the office, to say these are the guidelines and then with the side teams because they took the resources and they were able to incorporate it in their health education, in their play sessions…We took the hard copies and the nice things that we took there were in different languages. So if we were having a session with a Zulu speaking client, then we were able to share those hard copies as well…we also share with the health professional team that we work with in the facilities, so that they also understand what we are doing, they mustn’t be surprised that now we are doing this to the children. We show them that these are the guidelines from the workshop that we attended and these are what the recommendations are. (Gauteng, peri-urban 1)*

*We took from you and we made copies for all our centres, we gave it to them, laminated them for them and they put them up in the classrooms so that the teachers can be reminded that this is what they need to follow and what they can instil in the classes and the children. (Cape Town)*

*Fortunately, we do trainings every Fridays, so as soon as we received the, the whole memo we kind of integrated it with mental health also around children…we took the whole pamphlet and broke it down to trainings also on mental health around children, also on reporting incidents, and also on like developmental milestones for children. (Gauteng, urban)*

*After the training, we went to the facility managers…that’s where we shared the guidelines with them and they were so excited for the guideline…some of the posters, they took it and they paste it in each and every room where mother and child has been feeding, or where a mother comes in, can be able to identify and maybe be able to stimulate the child…They were actually excited and happy for us to bring those guidelines to them. (Gauteng, peri-urban 2)*


In a few of the focus groups, participants specifically mentioned how the guidelines helped them to encourage mothers to make toys for their children with what they have at home, rather than relying on being able to buy toys, which can often be expensive.


*Remember we have that tendency that in order for the child to play, we have to go to the shops and buy the toys. So what we did, we had to go back to them and show them other, I would say other skills to make toys for the child. We don’t have to go and buy because remember not all of us are working. Not all of us are getting enough money to buy the toys, so we have to teach them how to make toys, and then we have to teach them how to play with the children. (Gauteng, peri-urban 1)*


In terms of CBO’s efforts to disseminate the guidelines, participants mentioned some challenges, with COVID being one of the main challenges in 2020. Other challenges included the support needed by ECD practitioners for implementation, the high turn-over of community health workers, and the feasibility of some of the guidelines for families living in low-income settings.

#### 3.5.3. Responses of CBO Staff and End-Users

The focus group discussions indicated an overall positive response of end-users to the guidelines. Mostly, the guidelines seemed to be perceived as new knowledge that was generally welcomed by end-users, with CBOs being able to answer questions about the guidelines. This affirms the value of the dissemination workshops to enable CBO representatives to feel empowered in their own understanding of the guidelines so that they can share this with others. Some participants spoken about positive feedback from end-users on how meeting the guidelines had positively influenced the children in their care.


*Because the biggest questions were coming from parents…let’s say for instance you say no screen time, they’re asking then what must I do? What can I substitute the screen time with? Or what can I do with the child or what can the child do in the meantime, just so they don’t cry? Because then it’s not fair for you to take away the TV without introducing something else. (Gauteng, urban)*

*The professionals…after sharing with them those things, I would say the response was good, especially the PMTC (prevention of mother to child transmission) staff. Because they are the ones who are helping us with the children and all that. It also gave them…I would say a wake-up call on how relaxed we are with our children at home. (Gauteng, peri-urban 1)*

*The babies were not moving that much because…the parents would just put them in bed and not do anything with them throughout the day. So, we actually told them that they can move with the babies, do some activities, play with the babies and let them sit, not allow them to watch TV or any screens. And it was actually so great for them because they did not know what to do and we got some really good feedback from them was really good because they could see some changes as time went on. (KZN 1)*

*After they tried implementing what we taught them they started seeing results, like kids remembering things they had learned from outside of TV more than what they learned on TV. So I think they were quite happy with the information. (KZN 1)*


In some focus groups, participants mentioned that the guidelines are particularly important for young mothers (including adolescents) and that they had a positive response from grandmothers, since they often take on a primary caregiving role, especially for very young mothers.


*Most of the women are very young, they’re not able to talk to the children maybe to play with the children; they’re used to making them watch TV… In our district we have a lot of teenage pregnancy…They have to go back to school, so the grannies have to look after the child. So, we also teach the grannies how to look after the child when the mother is at school. (KZN 2)*

*People who were listening more to these more were more the grandparents who were there; I mean older ladies they will understand this information better than the younger ones. (Gauteng, urban)*


#### 3.5.4. Perceptions of “Woza, Mntwana” Song and Recommendations for Dissemination

Participants were overwhelmingly positive about the “Woza, Mntwana” song, and it was evident from some of their comments that the underlying messages of the song (e.g., connection between mothers and children) were picked up by participants. At some focus groups, women attending had brought their own baby with, and participants even commented on how the babies responded positively to the song when it was being played during the focus group.


*I can hear it playing on a radio, it is and it’s quite catchy with the chorus you know. I think I like the fact that there’s, what four, four five different languages. (Cape Town)*

*Very beautiful indeed. (Gauteng, urban)*

*I love the rhythm…And I love the message as well, and the fact that uh we’ll be able to be singing and playing uh along with the babies…It encourage the mother to… say to the child come closer baby, come closer…it’s a good sing along…I think if the mother can sing and dance with the child…even now the babies were concentrating on the song… Here is the baby singing (laughs)…Can you hear the baby singing? (Gauteng, peri-urban 2)*


Participants agreed that it could be used during various activities with young children, such as baby stimulation, getting babies to sleep, as a “pack up song” for children at ECD centres, or as an activity at ECD centres that could encourage development.


*So that’s where you can play with your child while listening to the song, and then you can… the baby can even fall asleep, you can sing it to the baby… yes and it helps uh, uh I think it will help a mother and a child to bond…It is the perfect song; you can sing along with the baby or play it when bathing the baby. (Gauteng, peri-urban 2)*

*The song is useful to our children especially in vocabulary, the children uh when they are singing they are developing their language…And another thing, they, they will be singing in a group; so when singing in a group they’ll, they’ll socialise with other children…so for me this song is very useful especially for our children. (Gauteng, peri-urban 1)*

*I would use the song in the morning as a morning ring just before the activities and maybe add a few dance moves so we can be active. (KZN 1)*


Some participants suggested that the song needed some actions or movements so that ECD practitioners and caregivers would know what to do with the song.


*They (children) would probably sing it, but it would be nice to have something that you could put actions to as well, which you could, that you could also do more. (Cape Town)*

*I think for them to sing it with me first would be really nice so they know how to dance to it so they know how to use it otherwise it’s pointless to give them something they don’t know how to use you know, I mean it’s a very simple song so there shouldn’t be anything complicated about it. (KZN 1)*


With regards to participants’ recommendations for dissemination, sharing the song via WhatsApp was the most popular suggestion, as this could easily be shared with a CBO’s end-users and networks. This could then be played in a range of settings using a Bluetooth speaker, which seemed to be easily accessible—more so than CD players. Keeping the file size small was seen as important, in order to limit data costs. Other suggestions were sharing it via USB (and playing it on a laptop); a CD or DVD that could, for example, be played in health facilities where mothers are waiting; sharing on social media; sending a YouTube link; or playing on the radio. Some participants suggested adding the song to existing resources for ECD centres.


*In our play group, so they usually… one would be bringing their own… oh, I know all of them actually brought their Bluetooth speakers and then whatever song they needed to do uh I would just help them download it and then circulate it on our WhatsApp group and that’s how they would get it. (Gauteng, urban)*

*I also think if you want it, it becomes like, for practitioners, like a nursery rhyme, they do it every day with the kids, the kids go out, go home and sing it all the way, so the same with the song, if they teach the children in class, they will go home and parents will listen to what they singing. (Cape Town)*


## 4. Discussion

This paper reports on the feasibility and acceptability of community-based workshops to disseminate the SA 24-h movement guidelines for birth to five years, the extent to which these guidelines were disseminated further, and the development of a song about these guidelines. The findings of the evaluation presented in this paper indicate that it was feasible to arrange community-based dissemination workshops, reaching a range of urban and rural settings around SA. The generally positive feedback from workshop attendees provides evidence of the acceptability of this dissemination approach, particularly in low-income settings. The workshop format seems to have been acceptable to CBOs, particularly since there was no charge for attending a workshop. This could be because in low-income settings (especially rural), training opportunities tend to be limited; therefore, offering a free training opportunity could have been well received, especially since it was covering something novel but still relevant to their scope of work.

The extent to which guidelines were shared by CBOs is promising; however, participants were honest about the challenges of sharing and implementing these guidelines. These findings echo what was already found in the stakeholder consultation process [7] in terms of other issues in SA that contend with healthy movement behaviours in early childhood. Without systemic changes, these competing priorities will likely always be a reality, especially in low-income settings, emphasising the need for a long-term view of changes in these behaviour in order to realise more sustainable impact. The positive response to the “Woza, Mntwana” song is encouraging for the further development of contextually relevant creative materials about the guidelines.

In terms of the programme theory for the dissemination of the guidelines, the purpose of these workshops was achieved: the workshops were successful at promoting the guidelines and communicating the importance and application of the guidelines. While a great deal of work is required to maintain contact with ECD CBOs that were involved in these workshops, it is evident that these workshops created a valuable opportunity to network with these CBOs. The findings of this evaluation are promising for the achievement of the short- and medium-term outcomes of the workshops, at least with CBOs that were most engaged in dissemination. Furthermore, while many of the conditions necessary for the workshop outcomes to be achieved were met, resources and support, along with the ability to prioritise the guidelines, are areas that require further work in order to achieve the long-term outcomes of the workshops.

These findings affirm the clear need for a community-based approach to the dissemination of behavioural and other health promotion guidelines in SA, as well as the value of linking with CBOs for dissemination efforts. This approach of working with ECD CBOs is especially helpful for maximising the reach to end-users, particularly caregivers and ECD practitioners. ECD CBOs are already reaching these end-users and having an existing relationship with community stakeholders can lead to them being a more trusted source of information. The importance of trust in engaging with communities around global health research has been highlighted and is especially relevant in where disparities can feed mistrust of academics [18], such as in SA with its persisting inequalities. In low-income SA communities, academics may not necessarily be seen as a trusted source for new information, hence the need for CBOs to help broker these trust relationships.

Establishing legitimacy is also a goal of community engagement [18], which is relevant for the dissemination of these guidelines. It could be argued that the inclusive stakeholder consultation process carried out for the development of the guidelines [7] helped lay the groundwork for this dissemination process. A number of key ECD CBOs were present at the consultation meeting with national government and non-government stakeholders, which helped to establishing the legitimacy of these guidelines with these CBOs (and government), forming the building blocks for a relationship of trust with CBOs and ultimately communities and end-users. Had the legitimacy of the guidelines not been properly established, it is possible that engaging CBOs in the dissemination process would have been much more difficult and time-consuming.

The findings of this evaluation underscore the importance of documenting guideline dissemination efforts to learn what works (and why) in different settings and how dissemination can be optimised for maximum reach and impact. The need for such evidence has been highlighted in a recent review on the dissemination of movement behaviour guidelines, although this review focussed on adults and predominantly high-income countries [15]. Furthermore, the findings of this study suggest that those developing behavioural guidelines across a range of settings need to keep dissemination in mind. This is not in terms of influencing the evidence that informs the guidelines but is to do with how key guideline messages can be distilled and packaged in acceptable formats that are feasible for dissemination. To have an even wider impact on behaviour change at the population level, incorporating the SA 24-h movement guidelines for birth to five years into health policy documents is a crucial long-term goal. This would not only maximise their reach but could potentially lend even more legitimacy to the guidelines. While efforts to engage the health sector at a policy level in SA have not yet been successful, it is possible that persevering with CBOs who are engaging the health sector at a community level could help nudge these guidelines up the health policy agenda in SA.

The limitation of having a workshop facilitator conducting the workshop evaluation is acknowledged (although this was necessary for logistical and financial reasons), as this may have hindered negative feedback from workshop participants. However, the questionnaires provided participants with an opportunity to provide honest feedback. It is also acknowledged that these workshops may have been “preaching to the converted” since they relied on positive responses from CBOs and their willingness to voluntarily attend a workshop. It is likely that these workshops attracted CBOs and individuals who were interested in learning about the guidelines and have therefore contributed to an overall positive perception about the guidelines and the workshops, which may not be reflective of all ECD CBOs.

With regards to the limitations of the follow-up focus groups, the response was low due to COVID-19 and the lockdown (despite the number of CBOs contacted to take part), which had a particularly negative impact on the ECD sector in SA. Many organisations were focussed on providing humanitarian relief (such as food parcels and feeding programmes) and supporting the safe re-opening of ECD centres in low-income settings, with no financial support from the government. This could have accounted for the lower response from CBOs working with ECD practitioners, since they had more pressing issues to deal with. We acknowledge that the CBOs who were willing to participate in a focus group may represent a biased, positive perspective since it is likely that CBOs who felt indifferent about the guidelines and workshops would not have responded. In addition, it is probable that CBOs who had not made any effort to further disseminate the guidelines after the workshop would not have been inclined to respond to the request to provide feedback on dissemination.

## 5. Conclusions

This work has produced novel findings of behavioural guideline dissemination in SA. Given the relative simplicity of the workshop format, it is reasonable to assume that these workshops could be replicated in other settings, particularly through partnership with CBOs. However, it would be important to ensure that those facilitating such workshops would have the ability to present material in a culturally appropriate manner and would be viewed as a trustworthy source of new information. These findings not only have relevance for other health promotion guidelines and for different age groups; they are also relevant for other LMICs and high-income countries where low-income communities would benefit from a community-based approach. This work can inform the dissemination of early years movement guidelines in other countries, including the WHO’s global guidelines [5]. Given the importance of setting young children on their best health and development trajectories, these findings can provide valuable guidance for the promotion of behaviours that contribute to optimal health and development in the early years.

## Figures and Tables

**Figure 1 ijerph-18-03071-f001:**
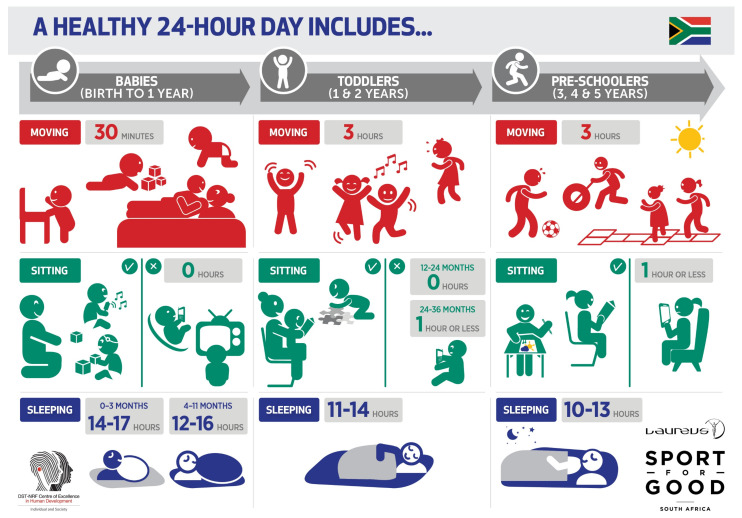
South Africa (SA) 24-h movement guidelines for birth to five years infographic [6].

**Figure 2 ijerph-18-03071-f002:**
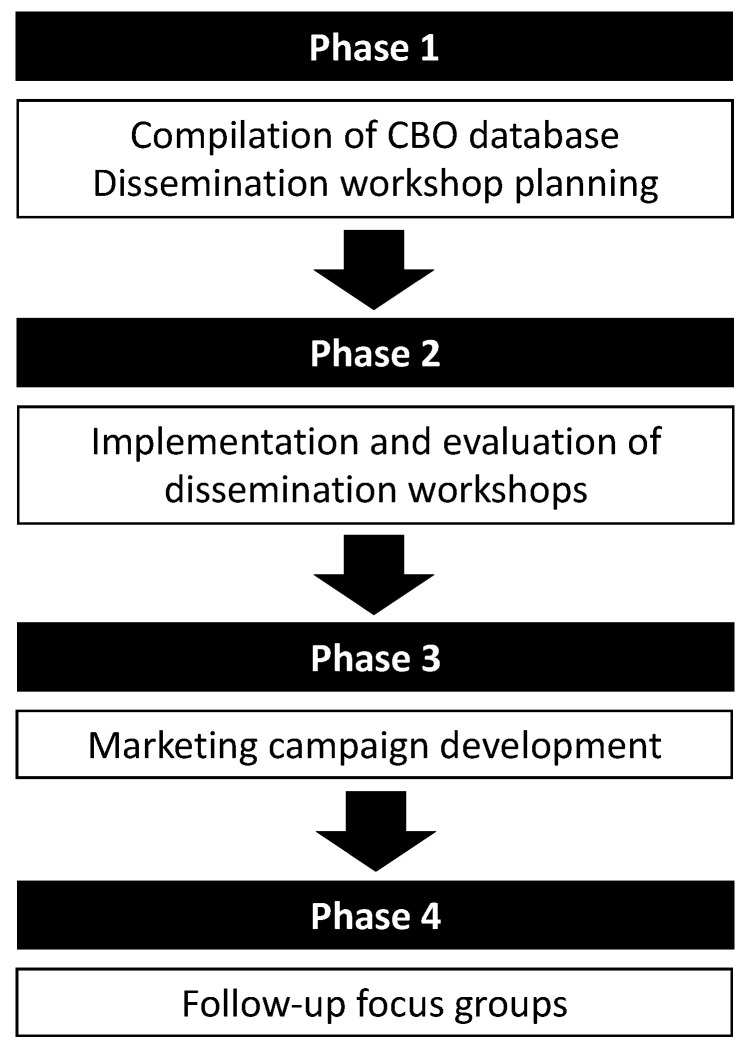
SA 24-h movement guidelines dissemination project phases. CBO, community-based organisations.

**Figure 3 ijerph-18-03071-f003:**
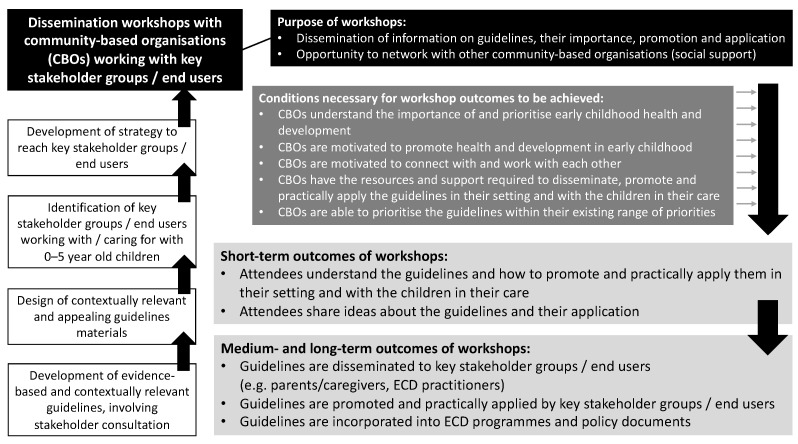
Guideline dissemination programme theory. ECD, early childhood development.

**Figure 4 ijerph-18-03071-f004:**
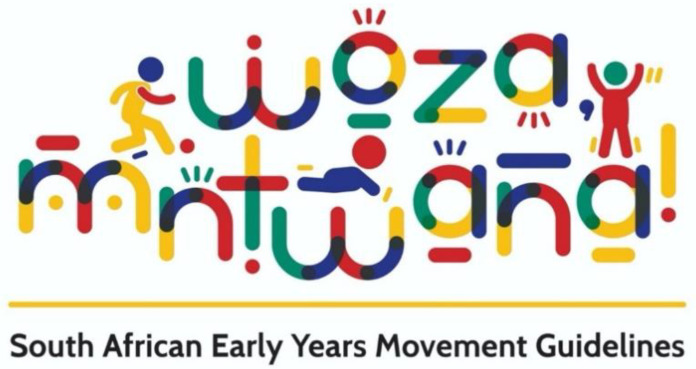
“Woza, Mntwana” campaign logo.

**Figure 5 ijerph-18-03071-f005:**
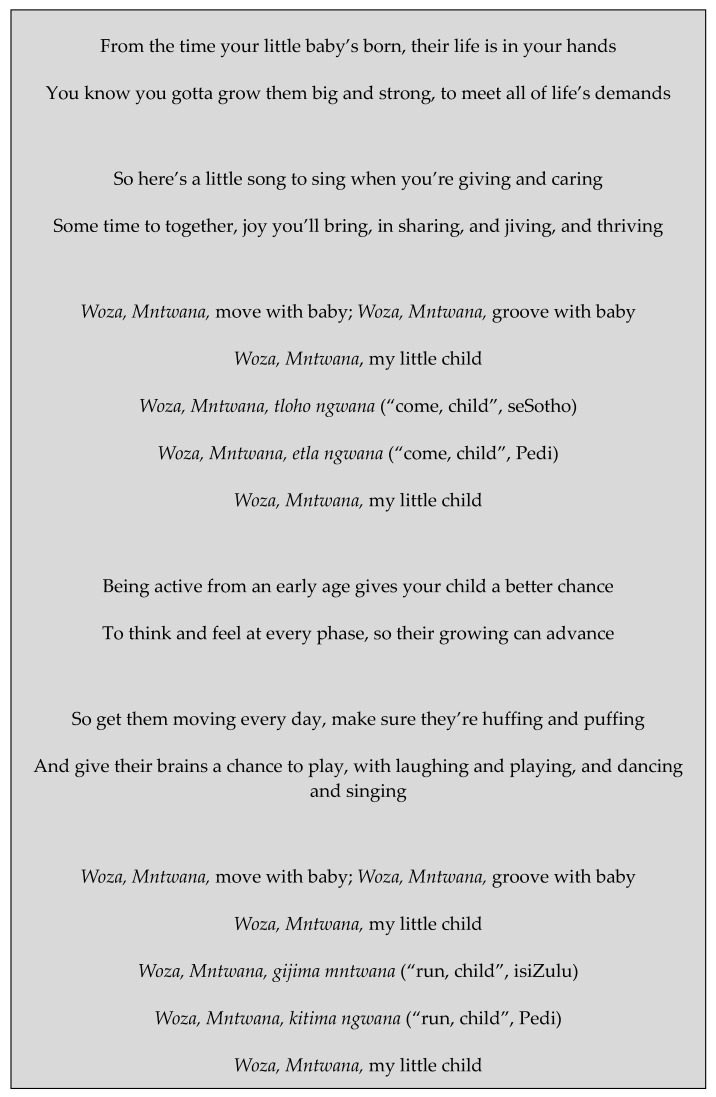
“Woza, Mntwana” lyrics.

**Table 1 ijerph-18-03071-t001:** Dissemination workshop format.

10 min	Arrivals and refreshments Provide name tags
5 min	Welcome and introductions
45 min	Presentation of guidelines Question time
30 min	Case studies (depending on time)
20 min	Verbal feedback from the group (group discussion)
10 min	Written feedback (evaluation form)
(120 min)	Workshop end

**Table 2 ijerph-18-03071-t002:** Dissemination workshop locations and attendees.

Workshop	Location	Attendees	Attendees (*n*)	Evaluation Forms (*n*, % Response)
1	Cape Town, Western Cape Province (urban)	CBO representatives, local government representatives, ECD practitioners, academics, students	30	30 (100%)
2	Cape Town, Western Cape Province (urban)	CBO representatives	11	8 (72.7%)
3	Durban, KwaZulu-Natal Province (urban)	CBO representatives	9	9 (100%)
4	Sweetwaters, KwaZulu-Natal Province (rural)	CBO representatives, community college representative	5	5 (100%)
5	Port Elizabeth, Eastern Cape Province (urban)	ECD practitioners, biokineticist (exercise therapist), student biokineticist	8	5 (6.3%)
6	Johannesburg, Gauteng Province (urban)	CBO representatives, community stakeholders	- *	- *
7	Johannesburg, Gauteng Province (urban)	CBO representatives, ECD practitioners	13	7 (53.8%)
8	Pretoria, Gauteng Province (urban)	CBO representatives, academics, students, ECD practitioners	101	78 (77.2%)
9	Polokwane, Limpopo Province (urban—small city)	CBO representatives	21	19 (90.5%)
10	Giyani, Limpopo Province (rural)	ECD practitioners	20	20 (100%)
11	Giyani, Limpopo Province (rural)	ECD practitioners	28	28 (100%)
12	Mbombela, Mpumalanga (urban—small city)	CBO representatives	7	2 (28.6)
13	Mbombela, Mpumalanga (urban—small city)	Provincial government representatives	14	14 (100%)
14	Bloemfontein, Free State Province (urban)	Academics, students	14	14 (100%)
15	Worcester, Western Cape Province (rural)	ECD practitioners	42	42 (100%)
		Total	323	281 (87%)

* Due to the format of workshop 6, attendee numbers were not recorded, and evaluation forms were not completed.

**Table 3 ijerph-18-03071-t003:** Workshop evaluation responses.

Workshop Evaluation Questions (*n* = 281)	Strongly Agree	Agree	Neutral	Disagree	Strongly Disagree
The workshop helped me to understand 24-h movement behaviours	54.8%	41.3%	3.2%	0%	0.7%
The workshop helped me to understand how to share the guidelines with others	51.2%	44.8%	3.2%	0.4%	0.4%
The workshop helped me to see the importance of healthy movement behaviours in young children	55.9%	40.2%	3.2%	0.4%	0.4%
I think that these guidelines are needed in South Africa	64.8%	33.1%	2.1%	0%	0%
I have the resources I need to promote the guidelines with the people I work with	37.7%	46.6%	13.2%	1.8%	0.7%
I have the support I need to promote the guidelines with the people I work with	38.4%	49.1%	11.4%	1.1%	0%
I feel confident that I can share these guidelines with the people I work with	54.8%	41.6%	3.2%	0.4%	0%
I would recommend this workshop to others who work with carers of children from birth to five years	60.5%	37.0%	1.1%	0.7%	0.7%
Open-ended responses to “Do you have any other feedback about the workshop on the guidelines?”					
“Discussion at the end was good. it is good for practitioners to discuss issues.” “Well-presented and clearly articulated as well as easy to follow and most importantly implement.” “Infographic was easy to understand.” “The importance of a routine according to age was novel.” “A very good workshop, sleep time was a revelation.” “I have gained a lot of knowledge. I was given ideas on how to implement these guidelines.” “I have gained a lot of knowledge and I will share with communities.” “I have been equipped with knowledge that I can pass on to parents.” “Parent workshops are needed.” “Practical examples needed, training of new parents and new teachers.” “Annual workshop would be beneficial.” “This is only the beginning; behavioural change is needed. Behavioural change needs multifaceted support.” “Media needs to be used to disseminate information. Parents need to be educated.”					

## Data Availability

The data presented in this study are available on request from the corresponding author. The data are not publicly available in order to protect the confidentiality of participants.

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
