# Peer review of "Evaluation of the Dissemination of the South African 24-Hour Movement Guidelines for Birth to 5 Years"

_ijerph, 2021, doi:10.3390/ijerph18063071_

Round 1
Reviewer 1 Report
Several basic aspects of an academic paper are missing in this paper. First of all, the academic problem and its significance are not clear. About the background of this study, the theoretical foundation is missing. About methods, the authors did not explain how the data was collected and analyzed. The Methods section only described the flow of the dissemination program, which should not be considered as the theory of the paper. Moreover, how the bias was considered and prevented especially when presenting the qualitative data was not explained. This problem is also reflected in the results part: the authors frequently used expressions such as "some attendees", "frequently discussed", etc., to show the tendency of feedbacks, however, failed to show how such frequencies have been identified. Moreover, the discussion part did not involve any discussions with previous studies. All these problems make this paper became very weak to be accepted as a qualified academic paper.
Author Response
Comment: Several basic aspects of an academic paper are missing in this paper. First of all, the academic problem and its significance are not clear.
Response: We apologise that the academic problem this paper addresses is not clear to Reviewer 1, but point out that the other reviewers identify the research to be interesting and that it deals with an “interesting and debated topic, [..] is well written and an accurate and valid methodological setup has been used”. A statement of an academic problem is not always relevant and/or appropriate in papers reporting on qualitative findings and implementation science, such as the paper we have submitted. At the end of the introduction, however, we state that ‘There is limited published evidence on the feasibility, acceptability, and effectiveness of movement guideline dissemination strategies, and focuses on adults’ (lines 109-110), and we articulate (lines 111-118) that the aims of the paper are to:
1) document the dissemination process the SA 24-hour movement guidelines for birth to five years, including the development of additional creative materials; and 2) report on:
- The feasibility of implementing the dissemination workshops;
- The acceptability of the dissemination workshops (and the guidelines) for different end-user groups; and
- The extent to which CBO representatives disseminated the guidelines to their staff and end-users.
We believe that the clear articulation of these aims, and the gap in research that this study is addressing (i.e. the significance of this paper) does not necessitate further clarification but would be happy to add further detail if the editor agrees this is necessary.
Comment: About the background of this study, the theoretical foundation is missing.
Response: Section 2.1 (line 129) presents the programme theory for the dissemination of the SA 24-hour movement guidelines for birth to five years. Given the pragmatic nature of this study, we do not believe that a theoretical foundation is required in the background of this study. Furthermore, in implementation science, a programme theory is typically provided. We have presented a background to the development of these guidelines, and the previous research that has informed our dissemination strategy. Furthermore, we have placed these guidelines in the South African context. While behaviour change theory is relevant the behaviour change that we hope to see as a result of the dissemination and implementation of these guidelines, we do not believe that behaviour change theories are specifically applicable to the dissemination of the guidelines, which is the focus of this paper. We are not aware of any particular theory that has been used for the dissemination of health promotion guidelines, but if the reviewer would like to recommend a theory they feel we have missed, we would be happy to consider its inclusion in the paper.
Comment: About methods, the authors did not explain how the data was collected and analyzed. The Methods section only described the flow of the dissemination program, which should not be considered as the theory of the paper.
Response: Figure 2 presents the flow of the dissemination process, as noted by the reviewer. However, Figure 2 was not intended to be considered as the theory of the paper, since Figure 3 presents the guidelines dissemination programme theory. The methods of each phase are presented in sections 2.2 – 2.5. These sections describe how data were collected for each phase, and where appropriate, data analysis was described. We have added additional information about analysis (changes highlighted in red):
In section 2.3 (lines 190-196):
‘Questionnaire responses were entered into Google Forms, and exported as an Excel spreadsheet for analysis. Frequencies were calculated for questionnaire responses in Excel, and are presented as percentages of responses who responded ‘strongly agree’. ‘agree’, ‘neutral’, ‘disagree’ or ‘strongly disagree’.. Open-ended questionnaire responses were grouped according to content, and group discussions were recorded transcribed verbatim. These open-ended responses and transcripts were thematically analysed by the lead author using a largely deductive approach.’
In section 2.5 (lines 243-247), the analysis is described:
‘a similar analytical approach was taken to identify the key themes from these focus group discussions, which included: 1) feedback on the workshops, 2) extent of dissemination to CBO staff and end-users, 3) responses of CBO staff and end-users, and 4) perceptions of Woza, Mntwana song, and recommendations for its dissemination.’
Comment: Moreover, how the bias was considered and prevented especially when presenting the qualitative data was not explained. This problem is also reflected in the results part: the authors frequently used expressions such as “some attendees”, “frequently discussed”, etc., to show the tendency of feedbacks, however, failed to show how such frequencies have been identified.
Response: We acknowledge that all research involves some degree of bias, and this has been mentioned in the limitations section of the Discussion (lines 820-842). We have acknowledged the potential bias of having the workshop facilitator conducting the workshop evaluation, the potential for positive responses, and that participants with more positive views may have been more inclined to be involved in the research. While it could be considered biased that the co-authors were involved in the evaluation, we believe this is an unavoidable bias that applies to most research studies – that the researchers who conducted the work are the ones who write up the findings for publication. If the reviewer feels that a particular bias has not been addressed, we would be happy to address this with further guidance.
When presenting qualitative findings in academic writing, it is common to use terms such as ‘some attendees’ and ‘frequently discussed’ to convey the extent to which certain topics were spoken about by participants. It is not appropriate to quantify the frequency of participants’ responses. This undermines the belief in qualitative research that all expressed opinions or perspectives of participants are of value, even if they are not mentioned frequently. We have therefore not provided numerical values in the qualitative results sections.
Comment: Moreover, the discussion part did not involve any discussions with previous studies.
Response: There have been no other studies that have assessed the dissemination of 24-hour movement guidelines (and physical activity guidelines in general) for the early years, or for any other age group of children. However, we do discuss and interpret the findings in relation to other relevant literature. We have included this additional sentence to highlight the paucity of evidence (lines 804-809):
‘The findings of this evaluation underscore the importance of documenting guideline dissemination efforts to learn what works (and why) in different settings, and how dissemination can be optimised for maximum reach and impact. The need for such evidence has been highlighted in a recent review on the dissemination of movement behaviour guidelines, although this review focussed on adults and predominantly high-income countries.’
Comment: All these problems make this paper became very weak to be accepted as a qualified academic paper.
Response: While we respect this reviewer’s opinion, we believe that the positive comments received from the 3 other reviewers indicate that our paper presents novel findings and makes an important contribution to the scientific literature.

Reviewer 2 Report
Dear Authors,
It has been a great pleasure to review this study.
Article deals with an interesting and debated topic, it is well written and an accurate and valid methodological setup has been used.
Congratulation for the video, well done.
Carefully revise punctuation throughout the manuscript.
I approve the publication of this paper after minor revision.
Title: ok
Abstract
Please, stay within the 200 words.
Keywords: ok
The discussion is too long, I suggest to shorten it
Line: 42: Please delate “in”
Line 64: check the link.
Line 72: Please, Not start the sentence with AND
Line 99: Delate [ref: Sonja’s Amagugu Asakhula paper].
Line 111: Delate “and”
Table
Table 2: Please, in the table 2 in line n. 5 attention to the % response
For better reading of the tables, I suggest spacing out the columns and not slurring words.
I suggest putting the Woza, Mntwana lyrics in the supplementary material.
Please, check the order to Back Matter
Author Response
Comment: It has been a great pleasure to review this study.
Article deals with an interesting and debated topic, it is well written and an accurate and valid methodological setup has been used.
Congratulation for the video, well done.
Carefully revise punctuation throughout the manuscript.
I approve the publication of this paper after minor revision.
Response: Thank you for these positive comments. We have reviewed the document to check punctuation.
Comment: Title: ok
Abstract
Please, stay within the 200 words.
Response: Apologies for this, the abstract was 201 words. We have removed 1 word to fit within the 200 word limit (line 20).
Comment: Keywords: ok
The discussion is too long, I suggest to shorten it
Response: We have made edits to shorten the discussion. Since these edits have been made throughout the discussion, we have not copied all edited sections into this response document, but revisions have been highlighted in the revised submission.
Comment: Line: 42: Please delate “in”
Response: Deleted.
Comment: Line 64: check the link.
Response: We have checked the link, and it is working (https://www.praekelt.org).
Comment: Line 72: Please, Not start the sentence with AND
Response: ‘And’ has been removed.
Comment: Line 99: Delate [ref: Sonja’s Amagugu Asakhula paper].
Response: Apologies for this oversight. This has been deleted, and replaced with a reference, since this paper has been accepted for publication.
Comment: Line 111: Delate “and”
Response: We have checked this; ‘and’ does not appear on line 111 (previous version). We have however made some other corrections to this text (lines 109-113):
‘There is limited published evidence on the feasibility, acceptability, and effectiveness of movement guideline dissemination strategies, and this evidence focuses on adults. This paper therefore aims to: 1) document the dissemination process of the SA 24-hour movement guidelines for birth to five years, including the development of additional creative materials; and 2) report on:’
Comment: Table
Table 2: Please, in the table 2 in line n. 5 attention to the % response
Response: If the reviewer is referring to workshop #5, the % is 6.25%, which has been rounded up to 6.3% for consistency (all rounded to 1 decimal place).
Comment: For better reading of the tables, I suggest spacing out the columns and not slurring words.
Response: We are not sure what the reviewer means by ‘slurring words’, but if it refers to carrying words over 2 lines, the formatting of the table (including spacing of columns) is created by the style template provided by the journal, and was not created by the authors. However, we have edited the table to ensure words do not run over 2 lines.
Comment: I suggest putting the Woza, Mntwana lyrics in the supplementary material.
Response: Given that a key part of the findings presented in this paper relate to the Woza, Mntwana song, we would request that these lyrics are kept as Figure 5 in the paper.
Comment: Please, check the order to Back Matter
Response: The order of the Back Matter is prescribed by the journal’s Word document template. We did not alter the order of the Back Matter as it appeared in the template.

Reviewer 3 Report
I find a very interesting research
The improvement proposals to consider are the following:
. Abstract: include aspects of the methodology (research design)
. Introduction: include a reference to other works that justify the procedure used for the dissemination and training of agents involved
. Methodology: include data on type of research; adjust the material and method section to subsections on participants, procedure (and here include the description of each phase); to specify more and better the types of instruments used (questionnaire and interviews) and include a section to specify the analysis and treatment of quantitative and qualitative data.
. Results: organize based on instruments and specify
. Discussion, in my opinion the data that appear in the debate does not really respond to a discussion since the results obtained are not contrasted with those of other previous research. It would be part of what this type of paper contributes once the conclusions are presented.
. Conclusions: Better organize the conclusions to respond to the research objectives
Author Response
Comment: I find a very interesting research
Response: Thank you for your positive feedback.
Comment: The improvement proposals to consider are the following:
. Abstract: include aspects of the methodology (research design)
Response: We would have liked to include further details about the methodology but given the 200 word limit are only able to include limited details about the methods to be able to clearly articulate the aims, state the dissemination strategy, and prioritise the presentation of results.
Comment: Introduction: include a reference to other works that justify the procedure used for the dissemination and training of agents involved
Response: Unfortunately there aren’t references for studies that have used this approach. We have mentioned in the introduction that South Africa is the first low- and middle-income country to develop 24-hour movement guidelines for young children, and that a community-based approach was considered a requirement for effective dissemination. High-income countries are able to rely to a large extent on online methods for dissemination, but we were not able to do this in South Africa (lines 55-59):
‘Given the wide range of intended end-users, dissemination would not be able to rely solely on online methods; any dissemination strategies would need to be optimised for low-income settings.’
Furthermore, we have stated the following:
‘there was limited precedent to draw on for the dissemination of health promotion guidelines in SA. The best known health promotion guidelines are the SA food-based dietary guidelines [11], but while these are widely supported by the Department of Health, dissemination at community level is limited [12].’ (lines 70-73)
‘There is limited published evidence on the feasibility, acceptability, and effectiveness of movement guideline dissemination strategies, and this evidence focuses on adults. [14]’ (lines 109-110)
In addition, feedback from the guideline consensus panel and stakeholders has guided the selection of this approach, which we believe is well justified for end-users of these guidelines in the South African setting:
‘Despite having little to draw on in terms of health promotion guideline dissemination, the consensus panel agreed that community-based dissemination of the movement guidelines was a priority.’ (lines 76-78)
‘Stakeholder consultation supported these decisions regarding these dissemination workshops [7]. (lines 92-93)
Comment: Methodology: include data on type of research;
Response: We have included the following description of the study to clarify the type of research:
‘Ethical approval for this mixed-methods study was obtained…’ (line 120)
Comment: adjust the material and method section to subsections on participants, procedure (and here include the description of each phase); to specify more and better the types of instruments used (questionnaire and interviews) and include a section to specify the analysis and treatment of quantitative and qualitative data.
Response: We appreciate that the current presentation does not follow standard practice for academic papers. While we have considered the suggestion to restructure the materials and methods section, given the phased nature of the study we believe that it could potentially be confusing to the reader to structure this section with the more traditional headings, and then explain the phases within each of these sections. We would therefore request to keep the headings as they are within the materials and methods section.
We did however edit the text on research instruments and data analysis are described in sections 2.3 and 2.5 to provide further clarity on the methods used. In section 2.3:
In section 2.3 (lines 190-196):
‘Questionnaire responses were entered into Google Forms, and exported as an Excel spreadsheet for analysis. Frequencies were calculated for questionnaire responses in Excel, and are presented as percentages of responses who responded ‘strongly agree’. ‘agree’, ‘neutral’, ‘disagree’ or ‘strongly disagree’.. Open-ended questionnaire responses were grouped according to content, and group discussions were recorded transcribed verbatim. These open-ended responses and transcripts were thematically analysed by the lead author using a largely deductive approach.’
In section 2.5 (lines 243-247), the analysis is described:
‘a similar analytical approach was taken to identify the key themes from these focus group discussions, which included: 1) feedback on the workshops, 2) extent of dissemination to CBO staff and end-users, 3) responses of CBO staff and end-users, and 4) perceptions of Woza, Mntwana song, and recommendations for its dissemination.’
Given the length of the paper, we have chosen not to include specific details of the questionnaire in section 2.3, since the questions are presented in Table 3. However, we have now included the focus group guide questions as supplementary material.
Comment: Results: organize based on instruments and specify
Response: For the same reason mentioned above, we believe that it is preferable for the flow of the paper and the readers’ understanding to present the results according to the project phases. We have specified the sections that present quantitative (line 269) and qualitative (line 284, and line 498) results.
Comment: Discussion, in my opinion the data that appear in the debate does not really respond to a discussion since the results obtained are not contrasted with those of other previous research. It would be part of what this type of paper contributes once the conclusions are presented.
Response: There have been no other studies that have assessed the dissemination of 24-hour movement guidelines for the early years, or for any other age group of children. However, we do reference other literature that is relevant to the interpretation of our findings. We have included this additional sentence to highlight the paucity of evidence (lines 804-809):
‘The findings of this evaluation underscore the importance of documenting guideline dissemination efforts to learn what works (and why) in different settings, and how dissemination can be optimised for maximum reach and impact. The need for such evidence has been highlighted in a recent review on the dissemination of movement behaviour guidelines, although this review focussed on adults and predominantly high-income countries.’
Comment: Conclusions: Better organize the conclusions to respond to the research objectives
Response: The discussion has addressed the aims of the paper, to: 1) document the dissemination process of the SA 24-hour movement guidelines for birth to five years, including the development of additional creative materials; and 2) report on the feasibility of implementing the dissemination workshops; the acceptability of the dissemination workshops (and the guidelines) for different end-user groups; and the extent to which CBO representatives disseminated the guidelines to their staff and end-users.
We believe that the conclusions at the end of the paper are an appropriate place to highlight the contribution of this study to the broader field of research on guideline dissemination, and how these findings might be applicable beyond South Africa. This is reflected in the current conclusion to the paper (lines 843-883):
‘This work has produced novel findings of behavioural guideline dissemination in SA. Given the relative simplicity of the workshop format, it is reasonable to assume that these workshops could be replicated in other settings, particularly through partnership with CBOs. However, it would be important to ensure that those facilitating such workshops would have the ability to present material in a culturally appropriate manner, and would be viewed as a trustworthy source of new information. These findings not only have relevance for other health promotion guidelines and for different age groups; they are also relevant for other LMICs and high-income countries where low-income communities would benefit from a community-based approach. This work can inform the dissemination of early years movement guidelines in other countries, including the WHO’s global guidelines [5]. Given the importance of setting young children on their best health and development trajectories, these findings can provide valuable guidance for the promotion of behaviours that contribute to optimal health and development in the early years.’
Given the length of the paper, and that another reviewer requested the discussion to be shortened, we believe that restating the aims, or explicitly referring back to these aims in the conclusion could be repetitive and would not necessarily add value to the paper.

Reviewer 4 Report
First of all I would like to share the need to carry out work like the one you present. They are necessary for the advancement of science in the field. The game is the basis in the early stages of life and takes on an important pedagogical value. The purpose of the manuscript is clear and consistent. The study has been an interesting read, however I have some questions and suggestions for the authors. The intervention carried out in the sectors that has been intervened seems really interesting to me, they are aspects that must go hand in hand, having the object of study as a common link. I would like to show that the instrument used is innovative and the documents that are attached to the manuscript help to understand the existing relationship in the matter. How was this awareness carried out for the participation of the participants? Because I understand that they have had a conversation with those responsible for the centers to choose them, by what means? Based on the results, orient the talks in the schools? Was that the same in all the schools? Did the same researcher do it? or it was several members who carried out this work. Don't you think that participation in one school and another may have influenced the results? I would like this appreciation to consist of work limitations. I would like to see the more detailed results section with respect to more specific aspects, since I consider that the data reflected in it is global and based on the research carried out, I believe that going even further into them will be relevant to know details from work. I also invite you to continue along this interesting line.Author Response
Comment: First of all I would like to share the need to carry out work like the one you present. They are necessary for the advancement of science in the field. The game is the basis in the early stages of life and takes on an important pedagogical value. The purpose of the manuscript is clear and consistent.
Response: Thank you for these positive comments.
Comment: The study has been an interesting read, however I have some questions and suggestions for the authors. The intervention carried out in the sectors that has been intervened seems really interesting to me, they are aspects that must go hand in hand, having the object of study as a common link. I would like to show that the instrument used is innovative and the documents that are attached to the manuscript help to understand the existing relationship in the matter.
Response: We are assuming that the reviewer is referring to the dissemination as the intervention, and that the instrument is the guidelines? We have stated that South Africa was the first low- and middle-income country to develop 24-hour movement guidelines for birth to five years (lines 41-42), so we believe that we have established the novelty of these guidelines. We trust that the figures and supplementary material (video) provided that display the guidelines are helpful.
Comment: How was this awareness carried out for the participation of the participants? Because I understand that they have had a conversation with those responsible for the centers to choose them, by what means?
Response: We apologise, but are not entirely sure we understand the reviewer’s questions. In section 2.2 (line 138), we explain how community-based organisations were contacted, how the workshops were planned, and how participants were invited to attend via email and telephone. Section 2.3 (line 173) explains how these workshops were carried out. If the reviewer could clarify what it unclear in these sections, we would be happy to address these points.
Comment: Based on the results, orient the talks in the schools? Was that the same in all the schools? Did the same researcher do it? or it was several members who carried out this work. Don't you think that participation in one school and another may have influenced the results?
Response: We are unsure why the reviewer refers to schools, and highlight that the research focussed on the feasibility and effectiveness of the dissemination strategy of movement guidelines for young children (under 5). Involvement of schools in the dissemination process was therefore irrelevant. While this research has relevance to early childhood development centres and involved early childhood development practitioners, there were no schools involved in this study as we anticipated that the workshop participants would lead further dissemination into the centres.
Comment: I would like this appreciation to consist of work limitations.
Response: We have included a limitations section in the Discussion (lines 803-825).
Comment: I would like to see the more detailed results section with respect to more specific aspects, since I consider that the data reflected in it is global and based on the research carried out, I believe that going even further into them will be relevant to know details from work. I also invite you to continue along this interesting line.
Response: Given that the results section already covers 11 pages, and presents results for each phase of the study, we are unsure as to how we would provide even more detailed results. The data presented are for South Africa (not global), but in the discussion we have pointed out the relevance of these findings to other low- and middle-income countries (lines 844-883):
‘This work has produced novel findings of behavioural guideline dissemination in SA. Given the relative simplicity of the workshop format, it is reasonable to assume that these workshops could be replicated in other settings, particularly through partnership with CBOs. However, it would be important to ensure that those facilitating such workshops would have the ability to present material in a culturally appropriate manner, and would be viewed as a trustworthy source of new information. These findings not only have relevance for other health promotion guidelines and for different age groups; they are also relevant for other LMICs and high-income countries where low-income communities would benefit from a community-based approach. This work can inform the dissemination of early years movement guidelines in other countries, including the WHO’s global guidelines [5]. Given the importance of setting young children on their best health and development trajectories, these findings can provide valuable guidance for the promotion of behaviours that contribute to optimal health and development in the early years.’
